# Changes in the Phytochemical and Bioactive Compounds and the Antioxidant Properties of Wolfberry during Vinegar Fermentation Processes

**DOI:** 10.3390/ijms232415839

**Published:** 2022-12-13

**Authors:** Ting Xia, Xiao Qiang, Beibei Geng, Xiaodong Zhang, Yiming Wang, Shaopeng Li, Yuan Meng, Yu Zheng, Min Wang

**Affiliations:** State Key Laboratory of Food Nutrition and Safety, Key Laboratory of Industrial Fermentation Microbiology, College of Biotechnology, Tianjin University of Science and Technology, Tianjin 300457, China

**Keywords:** wolfberry, vinegar, nutrition, bioactive compounds, antioxidant activity

## Abstract

Wolfberry (*Lycium barbarum* L.), as a kind of functional fruit, has various nutritional and bioactive components, which exhibit healthy benefits. However, wolfberry is not easy to preserve, and the intensive processing of wolfberry needs to be developed. In the present study, the changes in the phytochemical and bioactive compounds, as well as the antioxidant properties of wolfberry, were evaluated in the brewed processes. We found that the sugar contents were significantly decreased, and the total acids values were significantly increased during the fermentation processes. The sugar and fat contents were low in the wolfberry fruit vinegar after fermentation, which is of benefit to human health. In addition, amino acids were examined during the fermentation processes, and histidine, proline, and alanine were found to be the main amino acids in vinegar. The total phenolics and flavonoids contents were significantly increased by 29.4% and 65.7% after fermentation. 4-Hydroxy benzoic acid, 3-hydroxy cinnamic acid, and chlorogenic acid were the primary polyphenols in the wolfberry fruit vinegar. Moreover, the antioxidant activity of wolfberry fruit vinegar was significantly increased compared with that of wolfberry fruit after the fermentation processes. Polysaccharides and polyphenolics were strongly correlated with the antioxidant activity during the fermentation processes. The findings suggest that wolfberry fruit vinegar has a high antioxidant capability, and could be a beneficial food in the human diet.

## 1. Introduction

*Lycium barbarum* L. is known as wolfberry (goji berry), which belongs to the Solanaceae family. It has been used as a functional food and traditional Chinese medicine for thousands of years [1]. The wolfberry fruit planted in Zhongning is of a high quality, and is the only wolfberry cultivar that has been recorded in the Pharmacopoeia of the People’s Republic of China [2]. It has been reported that wolfberry has abundant nutrients and functional components, including polysaccharides, polyphenol, flavonoids, and amino acids [1,3]. The beneficial effects of wolfberry include antioxidation, hypoglycemic, and hypolipidemic properties, as well as anti-aging and the improvement of immunity [4].

Most of wolfberry fruits are produced into dried berries after harvest because the large yield of fruit is difficult to preserve [5]. The dried berries brown easily after exposure to air [6]. In order to maintain a better appearance quality and taste, deep processing of wolfberry fruits is necessary. Earlier studies have displayed that the processing products of wolfberry fruits are primarily wolfberry juice and wine, as well as being added into other food products, such as candies, cookies, and milk [7,8]. Brewed vinegar is an alternative strategy to promote the transformation of wolfberry, which needs further development.

Vinegar is used as a condiment worldwide, and is a type of functional food that is of benefit to the human body [9]. It is rich in nutrients and biological substances, including sugars, polyphenolic compounds, amino acids, organic acids, and melanoidins [10]. Many studies have reported that vinegar can prevent various diseases, such as diabetes, hypertension, infection, and inflammation [11]. The compositions and functions of vinegar are tightly related to the original materials and the fermentation processes [12]. Fruit vinegar is brewed through two steps, namely alcohol and acetic acid fermentation. In the alcohol fermentation process, yeast uses sugar to produce alcohol, and then the produced alcohol is used by acetic acid bacteria to produce acetic acid in the acetic acid fermentation process [13]. Fruit vinegar such as balsamic vinegar and sherry vinegar is mainly produced in European countries [14,15]. The raw materials of fruit vinegar are derived from grapes, apples, persimmons, and coconuts, among others [13,16]. However, the functional compounds during fermentation processes in fruit vinegar produced by wolfberry are still unclear.

In this study, fresh wolfberry was used as a raw material to ferment wolfberry fruit vinegar. The physicochemical parameters were determined during the fermentation processes. Meanwhile, nutritional and functional components were examined during the fermentation processes. Furthermore, the relationship between the antioxidant activity and functional components was revealed. The findings will provide an opportunity to make a functional fermentation food with raised economic added value.

## 2. Results and Discussion

### 2.1. Physicochemical Parameters of Wolfberry Fruit Vinegar during Fermentation

The changes in the physicochemical parameters during the fermentation processes are shown in Table 1. The pH values were not significantly altered between wolfberry fruit wine (WFW) and wolfberry fruit juice (WFJ) (*p* > 0.05). However, the pH value in wolfberry fruit vinegar (WFV) was 3.38 ± 0.08, which was significantly decreased compared with that in WFW (*p* < 0.05). Proper acidity can improve the taste and flavor of fruit vinegar [17]. In this study, there were no significant differences in the total acids between WFJ and WFW (*p* > 0.05). Nevertheless, the total acids value in WFV was 6.72 ± 0.12 g/100 mL, which was significantly increased compared with that in WFW (*p* < 0.05). It has been reported that acetic acid and other acids are produced during the acetic acid fermentation process [7]. Therefore, the increase in total acid value in WFV was caused by these acids.

In addition, the main non-volatile acids in fruit vinegars were lactic acid, citric acid, succinic acid, and malic acid, which modulated the irritation of acetic acid and provided a good dietary source of antioxidants [10,16]. In this study, the non-volatile acid contents were gradually increased during the fermentation process, which made the WFV taste softer. It is well-known that soluble solids are an important index to evaluate food quality [18]. The soluble solid content was significantly decreased to 8.42 ± 0.26 g/100 mL in WFW and 8.66 ± 0.28 g/100 mL in WFV, which was mainly due to the conversion from sugar into ethanol by active dry wine yeast [19]. The content of reducing sugar in WFJ sharply decreased from 12.08 ± 0.32 g/100 mL to 1.31 ± 0.11 g/100 mL in WFW, and then slightly increased to 1.42 ± 0.14 g/100 mL in WFV. This was mainly due to the reducing sugar used by the yeast during alcohol fermentation [20].

### 2.2. Nutritional Components of Wolfberry Fruit Vinegar during Fermentation

As shown in Table 2, the sugar content was decreased during alcohol fermentation. There was no significant difference between the WFW and WFV (*p* > 0.05). The decrease in sugar content during alcohol fermentation was related to the fact that sugar is decomposed by yeast and converted into ethanol [21]. The fat content values were very low in the fermentation stages, which were not significantly changed during the fermentation processes (*p* > 0.05). Low-sugar and low-fat diets are of benefit to human health [22]. These data reveal that the contents of sugar and fat were low in WFV.

Meanwhile, 15 amino acids were measured in WFJ (Table 3). The highest amount of amino acid in the WFJ sample was proline, followed by serine and histidine. Proline was also the predominant amino acid in WFW, followed by histidine. In WFV, the top two amino acids were histidine and proline. In general, histidine was significantly increased, while proline and alanine were significantly decreased during the fermentation processes (*p* < 0.05). Previous research has shown that proline was the amino acid in the greatest amount in Balsamic Vinegar of Modena (BVM), followed by alanine and threonine [23]. Another study reported that there were 16 amino acids in pear vinegars, and glutamic acid was the most dominant, followed by alanine and aspartic acid [24]. Amino acids in vinegar are mostly taken from raw materials and proteins are produced by microbial breakdown [25]. Therefore, the main amino acids in WFV were different from the fruit vinegars in other studies, which was may due to different raw materials and fermentation conditions [26].

### 2.3. Bioactive Components of Wolfberry Fruit Vinegar during Fermentation

As shown in Table 4, polysaccharide was the predominant bioactive component in WFJ. Wolfberry polysaccharide is a glycoconjugate, which has various biological functions, such as enhancing immunity, antioxidation, anti-inflammation, lowering blood glucose properties [3]. The polysaccharide content was significantly decreased to 7.26 ± 0.26 mg/mL in WFW, and then significantly rose to 8.94 ± 0.27 mg/mL in WFV (*p* < 0.05). Additionally, the betaine contents (2.68–2.94 mg/mL) showed no significant difference during the fermentation processes (*p* > 0.05), and betaine can adjust the taste and flavor of fruit vinegars [27]. The carotenoid content decreased from 0.58 ± 0.08 mg/mL to 0.42 ± 0.02 mg/mL along throughout the entire fermentation processes, which was mainly due to the oxidation reaction and degradation in the fermentation stages [28].

Moreover, the total phenolics content (TPC) and total flavonoids content (TFC) were the significant compounds in foods and plants, which have the capacity for free radical scavenging [29]. TPC was not significantly changed in WFW (*p* > 0.05), and was significantly increased to 2.42 ± 0.05 mg GAE/mL in WFV (*p* < 0.05). The increase in TPC during acetic fermentation was mainly attributed to combined polyphenols with sugars and organic acids to release free polyphenols [30]. TFC was enhanced to 1.63 ± 0.01 mg RE/mL in WFW (*p* < 0.05). There was no discernible difference in TFC contents between WFW and WFV (*p* > 0.05). Yang et al. reported that the TPC and TFC were 0.20 mg GAE/mL and 0.08 mg RE/mL in jujube juice, respectively. TPC and TFC were significantly increased to 0.28 mg GAE/mL and 0.09 mg RE/mL in jujube vinegar, respectively [31]. Wang et al. reported that the TPC was 0.53 mg GAE/mL during alcohol fermentation, and increased to 0.67 mg GAE/mL during acetic fermentation (*p* < 0.05). TFC was increased from 0.11 mg RE/mL to 0.13 mg RE/mL during the process of fermentation (*p* < 0.05) [16]. These results in previous studies indicated that TPC and TFC were increased during the fermentation processes, which were in line with our data.

### 2.4. Polyphenolic Compounds of Wolfberry Fruit Vinegar during Fermentation

Eighteen polyphenolic compounds were detected during the fermentation processes (Table 5). The top three polyphenolic compounds in WFJ were 4-hydroxy cinnamic acid, phloroglucinol, and 3-hydroxy cinnamic acid. Meanwhile, 4-hydroxyphenethyl alcohol was the predominant polyphenolic component in WFW, followed by phloroglucinol and 3-hydroxy cinnamic acid. In addition, 4-hydroxy benzoic acid was the highest polyphenolic in WFV, followed by 3-hydroxy cinnamic acid and chlorogenic acid. Moreover, the 4-hydroxyphenethyl alcohol content was increased first during alcohol fermentation (*p* < 0.05), and then decreased slightly during acetic acid fermentation (*p* > 0.05). Chlorogenic acid and 3-(4-hydroxy-3-methoxyphenyl) propionic acid gradually increased along throughout the entire fermentation (*p* < 0.05). Previous studies have shown that gallic acid, chlorogenic acid, and caffeic acid are the main polyphenolic compounds in the juice, wine, and vinegar of hawthorn [32]. Another study reported that tyrosol, protocatechuic acid, and gallic acid were the main polyphenolic compounds during the pomegranate vinegar fermentation processes [33]. Fruit raw materials are the primary source of polyphenolic components in fruit vinegar, and different fruit vinegars have different polyphenolic compounds [34]. In this study, the polyphenolic compounds in wolfberry vinegar were different from those in other fruit vinegars, mainly due to the different raw materials and production processes [10].

Furthermore, the polyphenolic compounds of WFJ, WFW, and WFV were characterized by PCA (Figure 1). The results showed that two PCs explained 97.1% of the variability (eigenvalues > 1), and the contribution rates of PC1 and PC2 were 59.7% and 37.4%, respectively. The polyphenolic compounds in the WFV sample were mainly found to be the positive for PC1, while those in the WFJ and WFW samples were found to be negative for PC1. In addition, the polyphenolic compounds in WFJ and WFV had positive values for PC2, while those in WFW were negative for PC2. PCA can be used to analyze and distinguish the different samples in the different fermentation stages, which had different distributions on the analysis graph [35]. The polyphenolic compounds in different samples during the fermentation process could be separated well, indicating the differentiated antioxidant characteristics.

### 2.5. Antioxidant Activities of Wolfberry Fruit Vinegar during Fermentation

As shown in Table 6, the antioxidant activity in WFW was lower than that in WFJ. This might be as a result of the massive reproduction of yeast, which lead to the loss of antioxidant components during alcohol fermentation [36]. Then, the antioxidant activity was significantly enhanced in WFV compared with that in WFW. It has been reported that the antioxidant activity of papaya vinegar was increased with the accumulation of acetic acid during the progress of fermentation [37]. Moreover, the values of the antioxidant activity in WFV were 27.87 ± 0.37 mM Trolox/L for 1,1-diphenyl-2-picrylhydrazyl (DPPH), 21.92 ± 0.28 mM Trolox/L for 2,2′-azinobis (3-ethlybenzothiazoline)-6-sulfonic acid (ABTS), and 8.76 ± 0.23 mM Trolox/L for the ferric ion reducing antioxidant power (FRAP), respectively, which were the highest in the fermentation processes. Özdemir et al. reported that the antioxidant activities were decreased in alcohol fermentation, and then raised during acetic acid fermentation by DPPH and FRAP [4]. Its antioxidant trend was similar to that in our study.

On the other hand, the correlation analysis between the functional ingredients and antioxidant activities is shown in Figure 2. The correlation coefficients between the contents of wolfberry polysaccharide and the antioxidant activities of DPPH and FRAP were 0.90, TPC and the antioxidant activities of DPPH were 0.94 in the fermentation process of WFV. These results indicate that wolfberry polysaccharides and total phenols exhibit a very strong correlation with the antioxidant activities. Several studies reported that the wolfberry polysaccharides and phenolic compounds showed antioxidant characteristics through scavenging the free radicals [29]. Collectively, polysaccharide and TPC were the main bioactive components, which contributed to the antioxidant activity in the fermentation processes.

## 3. Materials and Methods

### 3.1. Raw Materials and Chemicals

Wolfberry fruits were acquired from the Ningxia Zhongning Goji Industry Innovation Research Institute (Ningxia, China) in 2020. Active dry wine yeast was provided by Angel Yeast Co., Ltd. (Beijing, China). *A. pasteurianus* AC2005 was sourced from the Bioengineering College of Tianjin University of Science and Technology. DPPH, Folin–Ciocalteu reagent, rutin, and gallic acid standards were purchased from Sinopharm Chemical Reagent Co., Ltd. (Shanghai, China). The ABTS and FRAP assay kits were provided by the Beyotime Institute of Biotechnology (Shanghai, China).

### 3.2. Production of Wolfberry Fruit Vinegar

The fresh wolfberry fruits were washed and crushed into a homogenate using a pulping machine (Zhejiang Jinda Electric Machinery Co., Ltd., Zhejiang, China), and mixed with sterile drinking water (1:4, *w*/*v*). After filtering through a multi-layer filter cloth, 0.005% vitamin C and 1% citric acid were added to the filtrate to exert antioxidant activities. In the alcoholic fermentation process, the inoculation of the aforementioned activated yeast was 0.20% (10^6^ log CFU/mL) in the juice, and WFJ (1L) was fermented at 22 °C for 6 days in the incubator with a facultative anaerobic condition. The fermentation was terminated when there were no bubbles in the fermentation broth, and the obtained supernatant was WFW (about 850 mL). During the acetic acid fermentation process, *A. pasteurianus* were inoculated to acetic acid bacteria medium and cultured at 30 °C for 24 h to obtain acetic acid bacteria seed liquid. The WFW was inoculated with the acetic acid bacteria activated liquid of 10% (10^6^–10^7^ log CFU/mL) in the wine, and was fermented under aerobic conditions in a shaker at 180 r/min and 30 °C for 5 days, and the fermentation was stopped when the acetic acid content was stabilized (Figure 3). The obtained WFV was about 830 mL. To reduce the batch-to-batch variation of WFV, three batches of wolfberry fruits were collected in the same year.

### 3.3. Determination of the Physicochemical and Nutritional Components

The total acids, non-volatile acid, and reducing sugar were measured by acid–base titration, single boiling distillation units, and direct titration, in accordance with the Chinese National Standard methodology, respectively. A handheld refractometer was used to detect the content of the soluble solids. The sugar contents were determined using the phenol sulfuric acid method. In accordance with GB 5009.5-2016, the protein and fat were determined through the Kjeldahl method and Soxhlet extraction methodology, respectively.

### 3.4. Determination of the Amino Acids

The amino acids of the samples were determined using an amino acid automatic analyzer (S-433D; SYKAM, Munich, Germany). The analyzer functioned using a LCAK06/Na (4.6 mm × 150 mm) column. The mobile phase was an hydrochloric acid buffer with a 0.45 mL/min flow rate. The column oven temperature and the capacity of the injection were 30–70 °C and 50 μL, respectively. The ninhydrin reagents derivatized amino acids at 130 °C for 60 min, and were detected at 570 nm and 440 nm, respectively. The results were presented in milligrams per liter (mg/L) of samples.

### 3.5. Determination of the Functional Components

#### 3.5.1. Determination of the TPC

TPC was measured by the Folin–Ciocalteu technique [38]. Briefly, diluted samples were combined with Folin–Ciocalteu reagent (1:4, *v*/*v*). After being incubated for 5 min, 10% Na_2_CO_3_ and distilled water (1:5, *v*/*v*) were added to the samples in 96-well plates. The combination was kept in the dark for 2 h. The absorbance was determined at 765 nm using a microplate reader, and the data were presented in mg gallic acid equivalent per mL (mg GAE/mL).

#### 3.5.2. Determination of the TFC

TFC was measured using the colorimetric assay technique. Briefly, the diluted sample and 5% NaNO_2_ (2:1, *v*/*v*) were combined and maintained for 6 min in dark. Then, 20% NaOH and 5% Al(NO_3_)_3_ (4:1, *v*/*v*) were added and maintained for 6 min. Finally, the combined solution was placed in a 96-well plate and maintained for 15 min. The absorbance was detected at 510 nm using a microplate reader, and the data were presented in mg rutin equivalents per mL (mg RE/mL).

#### 3.5.3. Determination of the Polysaccharide

The polysaccharide content was determined using the phenol-sulfuric acid technique method. The samples were added into 5% phenol solution and concentrated sulfuric acid (1:5, *v*/*v*) at 40 °C water bath. After incubating for 10 min, the mixture, the blank of distilled water, and standard solution of glucose were placed in a 96-well plate. The absorbance was measured at 490 nm using a microplate reader.

#### 3.5.4. Determination of the Betaine

The betaine content in the samples was determined through a betaine assay kit. Briefly, the sample and 80% methanol (1:5, *v*/*v*) were combined, and then centrifuged at 10,000 rpm for 15 min. The mixture was maintained in a water bath at 60 °C for 30 min, and the supernatant was placed in a 70 °C water bath. Then, the betaine extract was diluted and measured using a microplate reader at 525 nm.

#### 3.5.5. Determination of the Carotenoids

The samples were mixed with solvents (n-hexane:acetone:toluene:ethanol = 10:7:7:6) and 40% KOH methanol solution (1:100:5, *v*/*v*/*v*). The mixture was placed in a 35 °C water bath for 12 h dark, and then extracted using diethyl ether. The extraction was added to deionized water and the underlying liquid was discarded following three uses. Finally, the extraction was passed through anhydrous sodium sulfate and diluted with diethyl ether. The absorbance was measured at 450 nm.

### 3.6. Determination of the Polyphenolic Compounds

The samples were centrifuged at 7104× *g* for 15 min, and the supernatant was ultrafiltered through a 10 kDa ultrafiltration cup (Shanghai Xuanyi Environmental Protection Technology Co., Ltd., Shanghai, China). The filtrate was purified using AB-8 microporous resin. Deionized water was used to remove impurities, and was eluted with 90% ethanol. Finally, the extract solution was evaporated and concentrated.

The polyphenolic compounds in the samples were detected through GC-MS with a Rtx-5MS column (30 m × 0.25 mm i.d., 0.25 μm). The flow rate of the purity helium was 1 mL/min, and the injector temperature was 300 °C. The oven temperature conditions were held at 80 °C for 2 min, and increased to 315 °C (5 °C/min) and maintained for 11 min. The temperature of the interface and ion source were 220 °C and 200 °C, respectively. The mass-to-charge ratio was 35–1000 m/z.

### 3.7. Antioxidant Activity Analyses

In the DPPH assay, the samples were diluted 30-fold with PBS. The diluted samples (20 μL) were mixed in 96-well plates with DPPH solution (180 μL). After 30 min at room temperature (in the dark), the samples were measured by microplate readers to read the absorbance at 517 nm and the results were represented as the trolox equivalent antioxidant capacity (TEAC).

In the ABTS assay, the samples were diluted 30-fold with PBS. The diluted samples (10 μL) and ABTS+ radical cation solution (200 μL) were combined in 96-well plates and incubated for 6 min (at room temperature). The samples were determined using a microplate reader to detect the absorbance at 734 nm, and the results were represented as TEAC.

In the FRAP assay, the samples were diluted 20-fold with PBS. The diluted samples (5 μL) and FRAP solution (180 μL), which were constituted by an acetate buffer (300 mM), FeCl_3_ (20 mM), and tripyridyltriazine (10 mM), and were placed in 96-well plates and incubated at 37 °C for 3–5 min (in the dark). The samples were detected using a microplate reader to measure the absorbance at 593 nm, and the results were represented as TEAC.

### 3.8. Statistical Analysis

The data were given as mean ± standard deviation (S.D., *n* = 3). Statistical analysis was performed using the GraphPad Prism 8.0.1 software (GraphPad Software Inc., San Diego, CA, USA), and SPSS 24.0 for Windows (SPSS Inc., Chicago, IL, USA). The Pearson’s correlation test was applied to the related correlation analyses. *p* < 0.05 suggested a significant difference.

## 4. Conclusions

In the study, it was found that goji berry fruit vinegar as a nutritious and functional product has a high antioxidant activity that could increase the economic added value. The changes in nutritional and bioactive compounds and the antioxidant activity of WFJ, WFW, and WFV were determined during the fermentation processes. Firstly, the phytochemical compounds were determined during the fermentation processes. The pH values were decreased and the total acids values were increased. Additionally, the sugar and fat contents were at low levels after fermentation. Histidine, proline, and alanine were the top three amino acids in WFV. TPC and TFC were significantly increased in the entire fermentation process. 4-Hydroxy benzoic acid, 3-hydroxy cinnamic acid, and chlorogenic acid were the important polyphenols compounds in WFV. Finally, the antioxidant activity of WFV was significantly enhanced after the fermentation processes. The anti-active ingredients of polysaccharide and TPC had a very strong correlation with the antioxidant activity in the fermentation process.

## Figures and Tables

**Figure 1 ijms-23-15839-f001:**
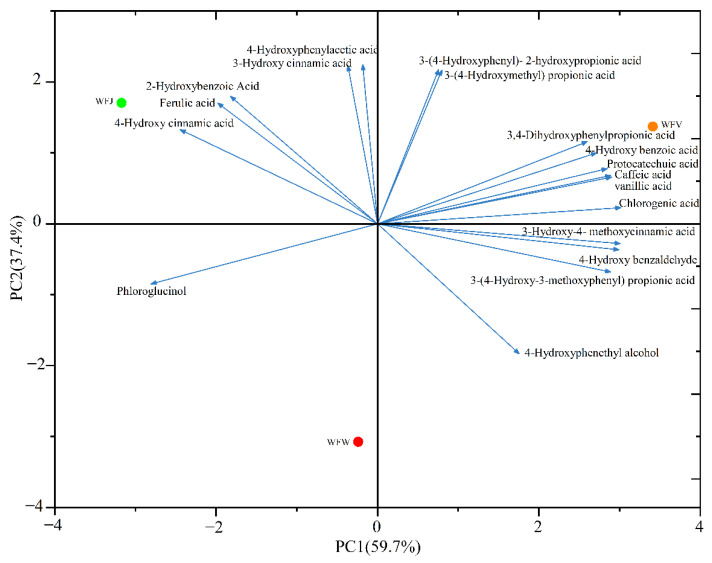
Principal component analysis of the polyphenolic compounds in the fermentation processes.

**Figure 2 ijms-23-15839-f002:**
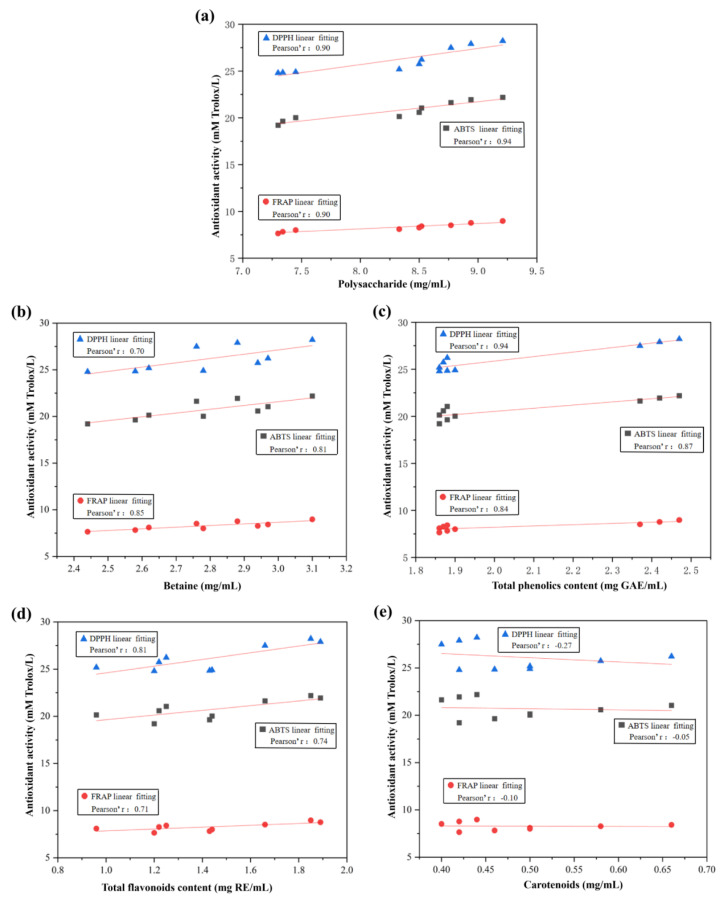
Correlation analysis of bioactive ingredients and antioxidant activity during the fermentation processes: (**a**) polysaccharide, (**b**) betaine (**c**) total phenolics content, (**d**) total flavonoids content, and (**e**) carotenoids.

**Figure 3 ijms-23-15839-f003:**
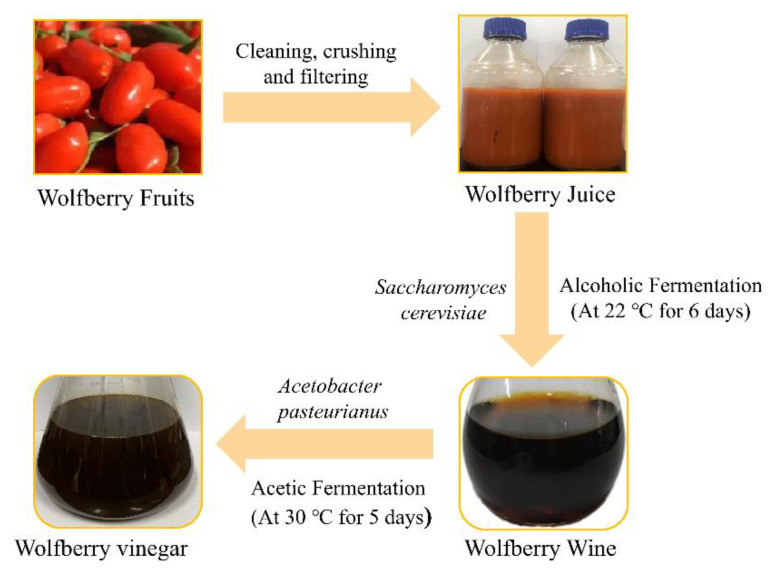
Vinegar production processes of wolfberry fruit.

**Table 1 ijms-23-15839-t001:** Contents of the physicochemical parameters in the fermentation processes of wolfberry fruit vinegar.

Products	pH Value	Total Acids (g/100 mL)	Non-Volatile Acid (g/100 mL)	Soluble Solids Content (g/100 mL)	Reducing Sugar (g/100 mL)
WFJ	4.48 ± 0.16 ^a^	0.56 ± 0.04 ^b^	0.42 ± 0.05 ^c^	16.26 ± 0.24 ^a^	12.08 ± 0.32 ^a^
WFW	4.20 ± 0.12 ^a^	0.60 ± 0.07 ^b^	0.56 ± 0.02 ^b^	8.42 ± 0.26 ^b^	1.31 ± 0.11 ^c^
WFV	3.38 ± 0.08 ^b^	6.72 ± 0.12 ^a^	0.84 ± 0.13 ^a^	8.66 ± 0.28 ^b^	1.42 ± 0.14 ^b^

WFJ: wolfberry fruit juice, WFW: wolfberry fruit wine, WFV: wolfberry fruit vinegar. a, b, c: values in the same row with different lowercase letters present statistically significant differences (*p* < 0.05).

**Table 2 ijms-23-15839-t002:** Contents of nutrients of wolfberry fruit vinegar in the fermentation processes.

Products	Sugar (g/100 mL)	Protein (g/100 mL)	Fat (g/100 mL)
WFJ	15.63 ± 0.24 ^a^	0.49 ± 0.08 ^a^	0.10 ± 0.02 ^a^
WFW	1.98 ± 0.15 ^b^	0.38 ± 0.06 ^a^	0.09 ± 0.04 ^a^
WFV	2.46 ± 0.12 ^b^	0.46 ± 0.04 ^a^	0.12 ± 0.02 ^a^

WFJ: wolfberry fruit juice, WFW: wolfberry fruit wine, WFV: wolfberry fruit vinegar. a, b: values in the same row with different lowercase letters present statistically significant differences (*p* < 0.05).

**Table 3 ijms-23-15839-t003:** Types and contents of amino acids in wolfberry fruit vinegar in the fermentation processes.

Compound	Amino Acid Content (mg/L)		
	WFJ	WFW	WFV
Proline	722.10 ± 18.72 ^a^	552.36 ± 26.88 ^b^	439.51 ± 18.34 ^c^
Serine	505.62 ± 18.92 ^a^	40.28 ± 5.42 ^b^	38.60 ± 3.53 ^b^
Histidine	353.41 ± 5.78 ^c^	433.79 ± 5.25 ^b^	543.65 ± 4.38 ^a^
Alanine	312.36 ± 16.97 ^a^	79.15 ± 12.65 ^b^	75.62 ± 8.97 ^b^
Aspartic acid	135.14 ± 4.38 ^a^	12.46 ± 2.45 ^b^	11.74 ± 2.50 ^b^
Arginine	107.59 ± 27.80 ^a^	68.87 ± 3.33 ^a^	66.54 ± 4.51 ^a^
Glutamic acid	73.59 ± 8.58 ^a^	50.53 ± 8.48 ^b^	43.38 ± 0.41 ^b^
Cysteine	54.78 ± 3.38 ^a^	42.04 ± 3.13 ^b^	45.03 ± 1.40 ^b^
Phenyllalanine	46.31 ± 14.73 ^a^	51.08 ± 15.29 ^a^	50.33 ± 9.82 ^a^
Lysine	40.80 ± 7.10 ^a^	6.83 ± 1.43 ^b^	4.89 ± 1.12 ^b^
Valine	36.91 ± 0.59 ^a^	-	-
Leucine	31.66 ± 2.81 ^c^	24.28 ± 0.36 ^b^	42.46 ± 1.75 ^a^
Threonine	29.50 ± 2.45 ^a^	21.85 ± 2.32 ^b^	25.41 ± 4.29 ^a b^
Isoleucine	8.70 ± 2.74 ^a^	-	11.51 ± 0.58 ^a^
Glycine	5.31 ± 0.41 ^b^	18.16 ± 4.15 ^a^	10.47 ± 2.53 ^a b^

WFJ: wolfberry fruit juice, WFW: wolfberry fruit wine, WFV: wolfberry fruit vinegar. a, b, c: values in the same row with different lowercase letters present statistically significant differences (*p* < 0.05). -: not detected.

**Table 4 ijms-23-15839-t004:** Contents of nutrients of wolfberry fruit vinegar in the fermentation processes.

Products	Polysaccharide (mg/mL)	Betaine (mg/mL)	Carotenoids (mg/mL)	TPC (mg GAE/mL)	TFC (mg RE/mL)
WFJ	8.58 ± 0.15 ^a^	2.94 ± 0.12 ^a^	0.58 ± 0.08 ^a^	1.87 ± 0.01 ^b^	1.02 ± 0.05 ^b^
WFW	7.26 ± 0.26 ^b^	2.68 ± 0.14 ^a^	0.46 ± 0.04 ^b^	1.88 ± 0.02 ^b^	1.63 ± 0.01 ^a^
WFV	8.94 ± 0.27 ^a^	2.88 ± 0.22 ^a^	0.42 ± 0.02 ^b^	2.42 ± 0.05 ^a^	1.69 ± 0.02 ^a^

WFJ: wolfberry fruit juice, WFW: wolfberry fruit wine, WFV: wolfberry fruit vinegar. TPC: Total phenolic compounds. TFC: Total flavonoids compounds. a, b: values in the same row with different lowercase letters present statistically significant differences (*p* < 0.05).

**Table 5 ijms-23-15839-t005:** Types and contents of polyphenols in wolfberry fruit vinegar in the fermentation processes.

No	Polyphenolic Compounds	Contents (mg/mL)		
		WFJ	WFW	WFV
1	4-Hydroxy cinnamic acid	0.408 ± 0.017 ^a^	0.116 ± 0.010 ^b^	0.131 ± 0.005 ^b^
2	4-Hydroxyphenethyl alcohol	0.004 ± 0.001 ^c^	0.337 ± 0.015 ^a^	0.216 ± 0.024 ^b^
3	Phloroglucinol	0.349 ± 0.005 ^a^	0.329 ± 0.008 ^a^	0.105 ± 0.004 ^b^
4	4-Hydroxy benzoic acid	0.100 ± 0.005 ^b^	0.079 ± 0.008 ^b^	0.317 ± 0.011 ^a^
5	3-Hydroxy cinnamic acid	0.339 ± 0.008 ^a^	0.195 ± 0.012 ^b^	0.311 ± 0.011 ^a^
6	Chlorogenic acid	0.026 ± 0.005 ^c^	0.096 ± 0.020 ^b^	0.222 ± 0.008 ^a^
7	3-(4-Hydroxy-3-methoxyphenyl) propionic acid	-	0.159 ± 0.007 ^b^	0.222 ± 0.008 ^a^
8	4-Hydroxy benzaldehyde	0.005 ± 0.001 ^c^	0.015 ± 0.002 ^b^	0.022 ± 0.001 ^a^
9	2-Hydroxybenzoic Acid	0.098 ± 0.008 ^a^	0.044 ± 0.003 ^c^	0.059 ± 0.002 ^b^
10	Ferulic acid	0.212 ± 0.001 ^a^	0.094 ± 0.004 ^c^	0.134 ± 0.005 ^b^
11	3-(4-Hydroxymethyl) propionic acid	0.141 ± 0.010 ^a b^	0.104 ± 0.009 ^b^	0.151 ± 0.005 ^a^
12	Caffeic acid	0.063 ± 0.009 ^b^	0.077 ± 0.005 ^b^	0.143 ± 0.005 ^a^
13	vanillic acid	0.042 ± 0.002 ^b^	0.051 ± 0.006 ^b^	0.097 ± 0.003 ^a^
14	3-(4-Hydroxyphenyl)-2-hydroxypropionic acid	0.034 ± 0.003 ^a^	0.018 ± 0.007 ^a^	0.038 ± 0.001 ^a^
15	4-Hydroxyphenylacetic acid	0.030 ± 0.002 ^a^	0.015 ± 0.002 ^b^	0.028 ± 0.001 ^a^
16	3,4-Dihydroxyphenylpropionic acid	0.014 ± 0.001 ^b^	0.014 ± 0.002 ^b^	0.027 ± 0.001 ^a^
17	3-Hydroxy-4-methoxycinnamic acid	0.006 ± 0.001 ^c^	0.037 ± 0.001 ^b^	0.062 ± 0.002 ^a^
18	Protocatechuic acid	0.005 ± 0.001 ^b^	0.007 ± 0.001 ^b^	0.022 ± 0.001 ^a^

WFJ: wolfberry fruit juice, WFW: wolfberry fruit wine, WFV: wolfberry fruit vinegar. Values in the same row with different superscripted alphabet are significantly different at *p* < 0.05. -: not detected.

**Table 6 ijms-23-15839-t006:** Antioxidant activity of wolfberry fruit vinegar in the fermentation processes.

Products	DPPH (mM Trolox/L)	ABTS (mM Trolox/L)	FRAP (mM Trolox/L)
WFJ	25.71 ± 0.52 ^b^	20.59 ± 0.46 ^b^	8.26 ± 0.16 ^b^
WFW	24.84 ± 0.06 ^c^	19.57 ± 0.33 ^c^	7.82 ± 0.18 ^c^
WFV	27.87 ± 0.37 ^a^	21.92 ± 0.28 ^a^	8.76 ± 0.23 ^a^

WFJ: wolfberry fruit juice, WFW: wolfberry fruit wine, WFV: wolfberry fruit vinegar. Values in the same row with different super-scripted alphabet are significantly different at *p* < 0.05.

## Data Availability

The authors confirm that the data supporting the findings of this study are available within the article.

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
