# Peer review of "Changes in the Phytochemical and Bioactive Compounds and the Antioxidant Properties of Wolfberry during Vinegar Fermentation Processes"

_ijms, 2022, doi:10.3390/ijms232415839_

Round 1

Reviewer 1 Report

Comments and Suggestions for Authors

Manuscript ID: ijms-2094546

Title: Phytochemical, bioactive compounds and antioxidant properties of fruit vinegar fermented from Wolfberry

This reseach revealed the changes of bioactive components and the antioxidant properties of wolfberry fruit vinegar in the fermentation processes. The results demonstrated that wolfberry fruit vinegar had unique flavor and higher antioxidant activities after fermentation, which could be a novel functional product to increase economic added value of wolfberry fruit. I recommend this manuscript for publication. However, there are some issues and concerns to address. Below are some of the more detailed comments:

1. Please provide the definition of “DPPH”, “ABTS”, and “FRAP” when first introducing them in section 2.5.

2. In section “Materials and Methods”, the fresh wolfberry fruits were crushed into homogenate. Whether water was added before alcohol fermentation?

3. Whether the functional ingredients were examined for the same batch in the fermentation process?

4. There are some typo, superscripts and subscripts, please double check them. For example, Na2CO3, NaNO2, 106 log CFU/mL.

5. Please check again reference [22], [24].

Reviewer 2 Report

1- In the tables, 3 products produced from goji fruit are mentioned, but only vinegar is mentioned in the title and summary. Why so?

2- It is not clear from the title and abstract that the study is a research article. It would be better to emphasize that it is a research article. So in this study, it should be said that we did and found the following.

3-Fruit vinegars could also be given comprehensively. The situation of fruit vinegars and how they were made could be explained.

4-I would like to see the information given in the conclusion section in the title, summary and introduction.

5-In general, a good topic has been chosen.

6-The product could have been called wolfberry and its other name(goji berry) could have been written in parentheses.

7-Line 318-319 … sentence is missing… product lost

‘In this study, changes of nutritional, bioactive compounds and antioxidant activity were determined during the fermentation processes. ‘

8- It is as if it was written at the end that should have been written at the beginning of the conclusion section.

If it were me, I would write this at the top of the conclusion section.

‘As a result of the study, it was found that goji berry fruit vinegar as a nutritious and functional product has high antioxidant activity to increase the economic added value.

9-I would like to see quotations from this study among the sources.

https://pubmed.ncbi.nlm.nih.gov/35204130/

10-For those who do not know the fruit, promotional photos of the fruit could also be added. Or vinegar production steps could be schematized.
